# Molecular Basis of Bile Acid-FXR-FGF15/19 Signaling Axis

**DOI:** 10.3390/ijms23116046

**Published:** 2022-05-27

**Authors:** Takeshi Katafuchi, Makoto Makishima

**Affiliations:** Division of Biochemistry, Department of Biomedical Science, Nihon University School of Medicine, Tokyo 173-8610, Japan; makishima.makoto@nihon-u.ac.jp

**Keywords:** bile acid, farnesoid X receptor, fibroblast growth factor 15/19

## Abstract

Bile acids (BAs) are a group of amphiphilic molecules consisting of a rigid steroid core attached to a hydroxyl group with a varying number, position, and orientation, and a hydrophilic side chain. While BAs act as detergents to solubilize lipophilic nutrients in the small intestine during digestion and absorption, they also act as hormones. Farnesoid X receptor (FXR) is a nuclear receptor that forms a heterodimer with retinoid X receptor α (RXRα), is activated by BAs in the enterohepatic circulation reabsorbed via transporters in the ileum and the colon, and plays a critical role in regulating gene expression involved in cholesterol, BA, and lipid metabolism in the liver. The FXR/RXRα heterodimer also exists in the distal ileum and regulates production of fibroblast growth factor (FGF) 15/FGF19, a hormone traveling via the enterohepatic circulation that activates hepatic FGF receptor 4 (FGFR4)-β-klotho receptor complex and regulates gene expression involved in cholesterol, BA, and lipid metabolism, as well as those regulating cell proliferation. Agonists for FXR and analogs for FGF15/19 are currently recognized as a promising therapeutic target for metabolic syndrome and cholestatic diseases.

## 1. Introduction

Bile acids (BAs) are amphipathic steroidal molecules produced mainly in the liver and partly in extrahepatic tissues, including macrophages and the brain. BAs are biosynthesized via a series of enzymatic reactions in the liver, stored in the gallbladder, and postprandially released into the small intestine as a detergent to facilitate digestion and absorption of fat-soluble nutrients. BA biosynthesis is also crucial for cholesterol catabolism and responsible for approximately 90% of daily cholesterol disposal in the body [1]. Hepatocytes synthesize primary BAs, such as cholic acid (CA), in both humans and mice, chenodeoxycholic acid (CDCA) in humans, and muricholic acid (MCA) in mice, and almost all primary BAs are conjugated with glycine (humans) or taurine (rodents) to increase solubility and stored in the gallbladder as bile salts. These primary BAs are converted to secondary BAs, such as deoxycholic (DCA) acid, lithocholic acid (LCA), and ursodeoxycholic acid (UDCA), by deconjugation and dehydroxylation enzymes in the enterobacteria during travel via the intestinal tract. BAs are mostly reabsorbed through various transporters expressed in the distal ileum. BAs are subsequently transported back to the liver via portal vein circulation as deconjugated primary BAs or secondary BAs and conjugated again for recycling. BAs in portal vein circulation also play multiple roles as endocrine hormones in the liver. To date, BAs have been shown to activate farnesoid X receptor (FXR) [2,3,4], vitamin D receptor [5], pregnane X receptor [6], Takeda G protein-coupled receptor 5 [7], sphingosine-1-phosphate receptor 2 [8], and M2 muscarinic acetylcholine receptor [9]. Among those receptors, FXR has been most extensively studied as it is the first-identified BA receptor, can be activated by different BA species, induces gene expression beneficial for our health, and has potential to develop novel interventions for liver disease.

FXR is also expressed in the distal ileum where BAs are actively reabsorbed and induce fibroblast growth factor (FGF) 15 (rodents) and FGF19 (non-rodents, including humans) expression, in addition to BA transporters. Furthermore, FGF15/19 is delivered to the liver via portal vein circulation and activates the receptor complex of FGF receptor 4 (FGFR4)-β-klotho (KLβ), an obligate co-receptor for FGF15/19 [10,11,12,13,14]. Collectively, BAs have two signaling pathways from the ileum to the liver, namely the direct pathway and the FGF15/19-dependent pathway. These signaling pathways are occasionally misinterpreted as stimulations with BAs and FGF15/19 elicit similar responses in hepatocytes. For example, both BAs and FGF15/19 increase levels of a small heterodimer partner (SHP) to suppress cytochrome P450 7A1 (CYP7A1) levels in the liver [15,16,17,18,19,20,21,22], of which differences between signaling pathways have received little research attention. The present review aims to explore FXR in the liver and ileum, focusing on the molecular basis of the FXR-FGF15/19 axis in BA signaling.

## 2. BA Biosynthesis

BAs are the major component in bile and contain a cholanic acid-backbone. BAs have both hydrophobic and hydrophilic properties and play a central role in emulsifying hydrophobic compounds, such as lipids, cholesterol, and fat-soluble vitamins to help absorption in the intestine [23]. There are over 30 different BA species in humans and rodents possessing multiple hydroxyl groups with different distribution and orientation, which determine potency of each BA [24,25]. BAs are biosynthesized mainly in the liver, called primary BAs, via oxidation, reduction, epimerization, translocation of the side chain, and conjugation of glycine or taurine to cholesterol catalyzed by 17 enzymes located in the endoplasmic reticulum, mitochondria, cytoplasm, and peroxisomes either through the primary pathway or alternative pathways [26]. Secreted BAs from the gallbladder are converted to different BA species, called secondary BAs, by gut microbiota during travel via the intestines [27].

### 2.1. Classic Pathway

The classic pathway is the principal cascade of BA synthesis (Figure 1), which is responsible for producing more than 90% of the BA pool in humans [28]. The first step is adding the 7α-hydroxyl group to cholesterol to produce 7α-hydroxycholesterol, which is performed by a cytochrome P450 (CYP) member, CYP family 7 subfamily A member 1 (CYP7A1), located in the liver microsome. This reaction is considered to be the rate-limiting step of this pathway and is also critical for cholesterol catabolism. In fact, both CYP7A1 deficiency in humans and CYP7A1 knockout mice exhibited hypercholesterolemia and reduced fecal BA levels [29,30]. This step is followed by catalyzation by 3-β-hydroxy-Δ^5^-C_27_-steroid dehydrogenase (HSD3B7) to produce 7α-hydroxy-4-cholesten-3-one (C4). Quantifying peripheral blood C4 levels is a non-invasive method for monitoring hepatic CYP7A1 activity, as peripheral blood C4 levels are strongly correlated with CYP7A1 enzymatic activity [31,32]. This compound receives 12α-oxidation in a CYP8B1-dependent manner, followed by side-chain oxidation by CYP27A1 and cleavage to produce CA. Alternatively, the compound can undergo aldo-keto reduction in an aldo-keto reductase (AKR1) D1- or AKR1C4-dependent manner, followed by side-chain oxidation with CYP27A1 and side-chain cleavage to produce CDCA. Finally, BA-CoA synthetase (BACS) and BA coenzyme A: amino acid N-acyltransferase (BAAT) catalyze conjugation of these BAs to glycine (humans) or taurine (rodents) [33]. CDCA is further converted to α-muricholic acid (MCA), followed with β-MCA by CYP2C70, which exists only in small rodents including mice [34], then finally undergoes taurine conjugation to be completed as tauro-β-MCA (t-β-MCA).

### 2.2. Alternative Pathway

As shown in Figure 1, the alternative pathways involve three different BA biosynthesis cascades [35]. Each pathway is primarily regulated by CYP27A1 [36], cholesterol 25-hydroxylase (CH25H), a non-cytochrome P450 enzyme [37], or CYP46A1 [38]. The CYP27A1-regulated pathway is initiated by converting cholesterol into (25R)-26-hydroxycholesterol by CYP27A1 and hydroxylation as the second reaction by CYP7B1 to produce (25R)-7α,26-dihydroxycholesterol, which is finally converted into CDCA in the liver [37]. These enzymes are also expressed in extrahepatic steroidogenic tissues including the adrenal gland for steroid hormone production and macrophages for oxysterol synthesis. Abundantly circulating oxysterols, such as 24-hydroxycholesterol, 25-hydroxycholesterol, and 27-hydroxycholesterol can be transported into the liver for conversion into CDCA and CA [37]. CH25H is expressed in the liver and converts cholesterol to 25-hydroxycholesterol, followed by conversion to 7α, 25-dihydroxycholesterol by CYP7B1. CYP46A1 (cholesterol 24-hydroxylase) is almost exclusively expressed in the brain and plays a critical role in cholesterol turnover, primarily converting cholesterol to 24S-hydroxycholesterol to release into circulation by passive diffusion across the blood–brain barrier [39]. 24S-hydroxycholesterol in the circulation is then associated with high-density lipoprotein (HDL) and low-density lipoprotein (LDL) [40], incorporated in the liver and metabolized into BAs [41,42]. Although *Cyp7a1* and *Cyp27a1* double-deficient mice exhibit a largely reduced BA pool size, they still retain a significant amount of BA species, including CA and CDCA, combined with similar expression patterns to WT mice for genes involved in BA regulation, synthesis, conjugation, and transport [43,44]. These data suggest that both CH25H and CYP46A1 significantly compensate for BA production in the absence of CYP7A1 and CYP27A1.

### 2.3. Metabolism of BA by Gut Microbiota

In the intestinal tract, primary BAs are converted into secondary BAs by the gut microbiota (Figure 2). In fact, eradicating gut microbiota by administering ampicillin significantly reduces DCA levels in the BA pool and fecal excretion of BA in humans [45,46]. Although DCA and LCA are the most abundant secondary BAs, approximately 50 secondary BA species have been detected in human feces [47]. Here, we mainly deal with DCA, LCA, ursocholic acid (UCA), and UDCA, the most abundant secondary BAs in both humans and mice, and several excellent reviews highlight the roles of gut microbiota in transforming other secondary BAs [27,48,49,50]. The first modification step involves deconjugation of glycine or taurine by bile salt hydrolases (BSHs) [51]. BSHs are found in almost all known major phyla of gut microbiota, including Bacteroidetes, Firmicutes, Actinobacteria, and Proteobacteria [52], most of which reside in the ileum and large intestine [27]. The 7α-hydroxy group is then removed by a series of enzymatic reactions from deconjugated CA and CDCA for DCA and LCA, respectively, in humans. The enzymes responsible for 7α-hydroxylation exist only in a few bacteria in *Clostridium* spp. and are named Bai enzymes due to their encoding by bile acid-inducible (*bai*) operon [27,53]. CA and CDCA are also converted to UCA and UDCA, respectively, from 7α-hydroxysteroid dehydrogenase (HSDH) and 7β-HSDH activities [27,48,54]. *Eggerthella lenta* strains contain 7α-HSDH and convert CA to 7-oxo-DCA, which is further converted to DCA by 7β-HSDH in *Clostridium* spp. [55]. *Eggerthella lenta* also possesses 3α- and one 3β-HSDH for converting DCA to isoDCA to reduce gut bacteria toxicity [56]. These HSDHs also convert LCA to 3-exoLCA and isoLCA, both of which modulate Th17 populations in mice [57,58].

### 2.4. Recycling of BA

BAs traveling via the intestinal tract are actively transported into the portal circulation via the apical sodium-bile acid transporter (ASBT), mainly in the apical brush border of enterocytes in the ileum and colon [59], whereby ASBT mutations can cause idiopathic intestinal disorder by primary BA malabsorption [60,61]. Approximately 95% of BAs reabsorbed into the body and only approximately 5% of remainders undergo fecal elimination [62]. Inside the enterocytes, ileal bile acid-binding protein (IBABP) facilitates BA transport from the apical to basolateral side of the cell [63]. IBABP deficiency indeed impairs apical to basolateral transport of BAs in ileal enterocytes [64]. Organic solute transporter (OST) α and β in the basolateral membrane of enterocytes transport BAs from enterocytes into portal vein circulation for delivery to the liver [63,65,66]. In the liver, Na^+^-taurocholate co-transporting polypeptide (NTCP) incorporates BAs from portal circulation into hepatocytes in a Na^+^-dependent manner [63,67,68,69], while organic anion-transporting polypeptide 1B1 (OATP1B1) and OATP1B3 recover BAs in a Na^+^-independent manner [63,70]. Reabsorbed BAs are simply conjugated again mostly with glycine in humans, while some secondary BAs, such as DCA, are 7α-hydroxylated again by CYP2A12, a non-human enzyme, prior to taurine conjugation in mice. These BAs are secreted into the canaliculus through ABC transporters for gallbladder delivery [71] and released into the duodenum with de novo synthesized BAs. Approximately 20% of the BA pool is occupied by DCA in human bile due to the absence of CYP2A12 [72], while only less than 1% of DCA is detected in the murine BA pool [73].

## 3. FXR

FXR is a nuclear receptor family consisting of 48 and 49 members in humans and rodents, respectively, and originally discovered by two independent groups. Seol et al. isolated two FXR splicing variants as the other part of retinoid X receptor alpha (RXRα) in the yeast two-hybrid screening library using human RXRα ligand-binding domain as bait [74]. Forman et al. found a novel orphan receptor by PCR screening using degenerate primers corresponding to the highly conserved DNA-binding domain of nuclear receptors in a rat cDNA library and was named according to its activation by farnesol metabolites [75]. In 1999, FXR was finally identified as a BA receptor by three independent groups [2,3,4]. Most natural BA species can somehow activate FXR and major BA activation is ranked in order of potency as CDCA > DCA > LCA > CA [76]. However, ASBT presence in the plasma membrane is necessary, particularly for primary BA species conjugated with glycine or taurine, since they show higher hydrophilicity than other unconjugated BAs and cannot cross the plasma membrane via passive transport [5]. In mice, tauro-β-MCA can antagonize intestinal FXR activity in mice [77,78], while natural human BA or its derivative equivalent to tauro-β-MCA has not yet been identified.

### 3.1. FXR Structure

FXR shares common structural features with other NRs. The amino acid sequence identity of FXR with other NRs reveals it is categorized as an NR1 (thyroid hormone receptor-like) subfamily member within the seven subfamilies (NR1-7) of the nuclear receptor superfamily. Other NR1 subfamily members include constitutive androgen receptor, liver X receptor α (LXRα) and β (LXRβ), peroxisome proliferator-activated receptor α (PPARα), β/δ (PPARβ/δ), and γ (PPARγ), retinoic acid receptor α (RARα), β (RARβ), and γ (RARγ), reverse Erb α (Rev-Erbα) and β (Rev-Erbβ), thyroid hormone receptor α (TRα) and β (TRβ), and vitamin D receptor (VDR). Among them, LXRs, PPARs, RARs, TRs, VDR, and FXR, form a functional heterodimer with RXRα with their DNA-binding element [79].

There are two FXR genes, FXRα (NR1H4) and FXRβ (NR1H5). FXRα contains four isoforms produced from the *Nr1h4* gene using either exon 1 (isoform 1 and 3) or exon 3 (isoform 2 and 4) for the different N-terminus and either includes (isoform 1 and 2) or excludes (isoform 3 and 4) an additional 12 bases resulting in insertion of four amino acids (=YMTG) in the hinge region [80,81]. Although the physiological role of this diversity is not completely characterized, significant differences were observed in tissue distribution of these isoforms and expression patterns of the four isoforms are involved in regulating FXR-dependent expression of target genes [82], BA and lipoprotein metabolism [83], as well as hepatic lipolysis and fatty acid metabolism [84]. FXRβ exists in human chromosomes as a pseudogene and the biological function of FXRβ is not well characterized, even in other species. In this review, we will only discuss FXRα, which hereafter is referred to simply as FXR.

FXR consists of N-terminal activation of the function-1 (AF-1) domain, the DNA-binding domain (DBD), the ligand-binding domain (LBD), the C-terminal ligand-dependent activation of function-2 (AF-2) domain, and the hinge region (H) located between DBD and LBD as a flexible linker (Figure 3). Although the functions of AF-1 domain in FXR are not completely understood, AF-1 domain in other NRs is structurally variable and is more susceptible to post-translational modifications, such as phosphorylation and SUMOylation for ligand-independent activity regulation [85]. The only evidence regarding functions of AF-1 in FXR identified so far is its interaction with β-Catenin and attenuating β-Catenin/TCF4 complex formation upon ligand binding [86].

The DBD of FXR consists of two α-helices (H1 and H2) and two Cys_4_ zinc fingers, and amino acid sequences of this domain are highly conserved among all NRs. This domain specifically recognizes the DNA motif called FXR responsive elements (FXREs). FXR in general forms a heterodimer with RXRα and recognizes inverted repeats of two FXREs separated by one nucleotide spacer (Figure 3B), called inverted repeat 1 (IR1) [75,87,88] or everted repeats separated by two or eight nucleotide spacers, called everted repeat 2 or 8 (ER2 or ER8) [89,90]. From genome-wide chromatin immunoprecipitation sequencing (ChIP-seq) studies on murine liver and ileum, and human primary hepatocytes have revealed that FXR binds to diverse FXRE architectures with consensus sequences of IR1: 5′-AGGTCANTGACCT-3′ (palindromic sequences are underlined) and ER2: 5′-TGACCTNNGGGTCA-3′ (everted sequences are underlined) [91,92]. A negative FXR responsive element has been identified as 5′-GATCCTTGAACTCT-3′ in the promoter region of apolipoprotein A-I (apoA-I), a major component of high-density lipoprotein, and expression of apoA-I is repressed following stimulation with synthetic FXR agonist GW4064 [88]. Patients with progressive familial cholestatic liver or bile duct-related cholestatic liver disease indeed exhibit reduced serum apoA-I concentrations. However, the detailed molecular mechanisms behind how the agonist-bound FXR reduces apoA-I transcription remains unidentified.

The LBD of FXR consists of multiple α-helices (H3-H12) and has a typical ligand-binding pocket (LBP) observed in most nuclear receptors. LBP volume is approximately 300–400 Å^3^ when binding to most agonists, while the volume expands up to 1081 Å^3^ when FXR binds to the antiparasitic drug ivermectin [93,94,95]. Amino acid residues composing the LBP are mostly hydrophobic for stabilizing interactions with the ligand, but also contain some polar residues for critical hydrogen bond formation for correct ligand orientation and selectivity. For example, the carboxyl group on the side chain and the hydroxyl group on carbon-3 (C3) of 6-ECDCA form a hydrogen bond with Arg328 on H5 and His444 on H10/11 of FXR, respectively, for correct orientation. Interestingly, the hydroxyl group on C3 of 24(S),25-epoxycholesterol forms a hydrogen bond with Arg319 on H5 of LXRβ and this difference makes orientation of both ligands completely opposite to their receptors, despite the structural similarity between these ligands [96,97,98]. Upon agonist binding, FXR is allowed to interact simultaneously with two coactivators through intermolecular contact with the LXXLL sequence motif by altering the position of H12 in the AF2 domain. The ligand sensor assay using surface plasmon resonance showed that LBD in a liganded state with CDCA or its conjugates increased affinity to a peptide of coactivator SRC1 containing LXXLL sequence motif approximately ten times higher than an unliganded state [99]. When FXR binds to an antagonist, such as ivermectin, H12 increased its flexibility and dynamics to become invisible in the crystal structure, and only deformed H11 associated with NCoR peptide was observed.

### 3.2. Liver- and Ileum-Specific Function of FXR

FXR is widely distributed in tissues, such as liver, intestines, kidney, and adrenal gland. In the kidney, FXR is abundantly expressed in the proximal tubules and induces aquaporin 2 levels to decrease urine osmolarity [100], playing a beneficial role against acute kidney injury [101]. In the adrenal gland, FXR contributes to glucocorticoid synthesis [102]. However, the detailed functions of FXR in these tissues has not been completely characterized, thus we focus on hepatic and intestinal functions.

BA biosynthesis is an important part of cholesterol catabolism and impaired regulation of BA metabolism can disrupt cholesterol metabolism, resulting in serious health damage. Indeed, FXR knockout mice exhibit elevated serum BA, triglycerides, total cholesterol, LDL, and HDL, as well as increased hepatic cholesterol and triglycerides, particularly in mice on high-cholesterol diets [103]. Based on the fact that active FXR reduces *Cyp7a1* expression and subsequent CYP7A1-dependent conversion from cholesterol to BA, phenotypes observed in FXR knockout mice appear to contradict our expectations, except for elevated BA levels. The hypercholesterolemia is actually explained by evidence that FXR activation inhibited phosphorylation levels of jun N-terminal kinase (JNK) to elevate mRNA levels of ABCG5, ABCG8, and scavenger receptor B1 (SR-B1) levels through HNF4α elevation, resulting in increased HDL uptake into the liver [104,105]. Regarding hyperlipidemia, FXR overexpression suppresses mRNA levels of SREBP-1c, PEPCK, and G6Pase in the liver to reduce plasma free fatty acid levels to subsequently improve insulin resistance in *db*/*db* mice [105]. FXR is thus considered the pharmaceutical target for developing drugs for non-alcoholic steatohepatitis (NASH) and non-alcoholic fatty liver disease (NAFLD), and synthetic agonists, such as obeticholic acid, tropifexor, cilofexor, EDP-305, and MET-409 have been developed and used in clinical practice or under clinical trials [106,107,108].

FXR also plays a critical role in the ileum to regulate BA transport from the enterocyte to enterohepatic circulation by increasing IBABP [109] and OSTα/β [110] levels. FXR activation instead suppresses BA absorption from the intestinal tract to the enterocytes by reducing ASBT levels on the apical surface via elevated SHP expression [111,112]. One of the most physiologically significant ileal gene targets of FXR is *Fgf15/FGF19* and the detailed mechanism how FXR regulates gene expression will be discussed below. Genome-wide analysis of FXR binding in mouse intestine and liver identified tissue-specific FXR binding sites on promoter regions. For example, multiple FXR binding sites in *Slc51a* (Ostα) were found only in the intestine, but not the liver, while a FXR binding site in *Slc10a1* (NTCP) was detected only in the liver, but not the intestine [91]. These data demonstrate that the activity of FXR is regulated in a coordinated manner for tissue-specific functions.

## 4. Regulation, Production and Biological Function of FGF15/19

FGF15/19 is an endocrine hormone secreted from the distal ileum and its production is critically controlled by ileal FXR activity and regulated by a variety of BA species with varying concentrations in the intestinal tract [108,113]. FGF15/19 regulate de novo BA synthesis, cholesterol catabolism, glycogen synthesis, fatty acid metabolism and regeneration in the liver, skeletal muscle mass, and appetite in the brain (Figure 4). The mechanism behind FGF15/19 production in the ileum and biological action to the liver are summarized in Figure 5. The FGF19 analogs lacking its mitogenic activity are also considered as a promising candidate for hepatic disorders, such as NASH and NAFLD [114,115,116].

### 4.1. Discovery of FGF15/19

FGF15 was originally discovered by the representational difference analysis in a murine fibroblast cell line NIH3T3 transfected with either an empty vector or vector containing cDNA of E2A-Pbx1b, a fusion protein with pre-B cell leukemia containing the t(1;19) chromosome translocation [117,118]. The most striking feature in the structure of FGF15 was that the predicted amino acid sequence has a putative signal peptide, which does not exist in the canonical FGF family members [119]. The later discovery of FGF21 and FGF23 demonstrated that these FGFs share structural features with FGF15, such as a signal peptide and a short heparin binding domain, and are currently divided into a hormone-like subfamily within the FGF superfamily [120,121]. FGF19 was found by searching a human expressed sequence tag database [122]. Although different numbers are assigned, as a result of the relatively low amino acid sequence identity (approx. 50%), FGF19 was soon identified as a human orthologue of FGF15. In fact, both human *FGF19* and mouse *Fgf15* genes form a syntenic cluster with two other FGF genes, *FGF3/Fgf3* and *FGF4/Fgf4* in their genome [123].

Earlier research on the biological function of FGF15/19 mainly revealed that FGF15/19 is expressed in the fetal brain, is responsible for developing the nervous system and sensory organs, and is critical for survival during late embryonic development [118,124,125,126,127]. The first evidence on the biological function of FGF15/19 in hepatic physiology and pathology was discovered in human FGF19 transgenic mice. These FGF19 transgenic mice exhibited increased proliferation of pericentral hepatocytes and later development of hepatocellular carcinoma, mediated through FGF receptor 4 (FGFR4) [128]. They also exhibited reduced fat mass due to increased energy expenditure, which is accompanied with reduced expression of acetyl-CoA decarboxylase 2 in the liver [129]. Soon after this discovery, FGF19 was found to suppress *CYP7A1* expression in primary human hepatocytes and mouse liver through a c-JNK pathway. Further studies then identified a synthetic agonist FXR GW4064 [130] and natural agonist CDCA robustly and dose-dependently increased FGF19 expression through the FXR-RXRα heterodimer on their binding element IR1 located on intron 2 in primary human hepatocytes [11], despite later investigations clarifying these compounds do not induce hepatic FGF15/19 expression in vivo. The milestone discovery on the biological function of FGF15/19 in the liver was reported by Inagaki et al., showing that expression of *Fgf15* was abundantly detected in the ileum and strongly induced by oral administration of either CA or GW4064, whereby subsequent suppression of *Cyp7a1* levels was observed in the mouse liver [18]. They also observed the *Fgf15*- or *Fgfr4*-deficiency diminished *Cyp7a1* suppression levels by agonist administration. Later studies revealed the presence of KLβ in the liver as an obligatory co-receptor for FGF15/19. Collectively, FGF15/19 was discovered as a hormone for signaling from the ileum to the liver to regenerate and facilitate feedback regulation of BA synthesis.

### 4.2. Molecular Biological Basis for FXR-Dependent Regulation of FGF15/19

Expression levels of *Fgf15/FGF19* are principally regulated by FXR. Although FXR is highly expressed in the liver, kidney, and small and large intestines, *Fgf15/FGF19* are expressed the most in the ileum, slightly higher than the detection limit in the jejunum and colon, and undetectable in the liver and kidney. The mechanism underlying this tissue-selective expression of *Fgf15* remains unknown. *Fgf15/FGF19* genes have a typical IR1 element for binding the FXR-RXRα complex located between exon 2 and exon 3, and the electrophoretic mobility shift assay and luciferase reporter assay show this element is functional in BA-dependent regulation [18,131]. Disrupting FXR in the ileum completely abolished responsiveness of *Fgf15* expression upon BA stimulation [132]. However, FXR is not essential for basal expression of *Fgf15*, since the *Fxr*-disrupted ileum still expressed minimum, but significant, *Fgf15* levels [132]. Regarding the effect of natural antagonist tauro-β-MCA on *Fgf15* expression, *Cyp2c20*-deficient mice were previously used to remove β-MCA production to reveal basal *Fgf15* mRNA levels and responsiveness to FXR agonists were normal [133]. Furthermore, mice lacking both *Cyp2a12* and *Cyp2c70* genes to humanize BA metabolism exhibited no change in ileal *Fgf15* expression levels, although significant elevation was observed in some FXR-regulated genes, such as *Cyp3a11*, *Mdr1a*, *Srebf1*, and *Abca1* [134]. These data suggest that tauro-β-MCA has little effect on *Fgf15* expression in vivo.

Expression levels of *Fgf15* varied depending on intestinal microbiota conditions, whereby primary BAs were converted to secondary BAs. First, the eradication of mouse intestinal microbiota with antibiotic treatment greatly reduced ileal *Fgf15* levels, which was accompanied by deconjugated primary BAs, such as CA and β-MCA, and secondary BAs, such as DCA and HDCA, to faint levels in the BA pool [46,135,136]. Oral administration of CA to antibiotic-treated mice greatly elevated ileal *Fgf15* expression levels while tauro-CA did not, suggesting BA deconjugation by gut microbiota is critical for BA-induced *Fgf15* production [136]. The reduced *Fgf15* expression by antibiotic treatment was largely increased following subsequent colonization of human or mouse microbiota in the mouse ileum [137].

### 4.3. Interaction of FGF15/19 with FGFR4-KLβ Heterodimer

Expression of FGFR4 was observed mainly in the adrenal gland, kidney, liver, and lung, while KLβ mRNA was highly detected in adipose tissues and the liver [13,138]. Therefore, the primary target tissue of FGF15/19 is the liver, where FGFR4 is the predominant isoform of all FGFRs [138]. The potency of FGF19 on FGFR4 and KLβ ectopically expressed in HEK293 cells were evaluated by phosphorylation levels of FGFR substrate 2 (FRS2) and ERK1/2, and those were far greater when FGFR4 and KLβ were co-expressed compared with FGFR4 expression alone [12,13,14]. Furthermore, global or liver-specific KLβ deficiency not only abolished *Cyp7a1* repression in the liver after administering human FGF19 or murine FGF15, but also increased basal *Cyp7a1* levels due to the lack of endogenous FGF15 signaling [139,140,141]. The *Fgfr4* knockout mice or FGFR4 antisense oligo-treated mice also showed elevated expression of *Cyp7a1* levels [142,143], revealing that both KLβ and FGFR4 are critical for FGF15/19-dependent *Cyp7a1* suppression in the liver. Interestingly, FGF15 plasma concentrations in liver-specific KLβ-deficient mice were almost 30-fold higher than WT, suggesting that KLβ plays a central role in clearing circulating FGF15/19 [141].

The 3D structure of the FGF23-FGFR1c complex determined by X-ray crystallography revealed it requires an obligatory interaction with KLα and heparan sulfate (HS) for a 1:1:1:1 assembly for signaling [144]. Based on these data and biochemical evidence of FGF19 signaling accumulated in the past [10,11,12,13,14], FGF15/19-FGFR4 also requires interaction with KLβ and HS for signaling with a 1:1:1:1 ratio [145,146]. Furthermore, FGF19 mutants with reduced dimerization potential elicit lower mitogenic activity accompanied with lower induced ERK1/2 phosphorylation, whereas hepatic *Cyp7a1* expression reduced to similar levels as liver treated with WT [145,147,148]. These results suggest dimerization of the FGF15/19-FGFR4-KLβ-HS complex is required for eliciting mitogenic activity of FGF19, whereas dimerization is dispensable for FGF19-mediated *Cyp7a1* repression in the liver. Conversely, FGF15 induces weaker mitogenesis with ERK1/2 phosphorylation to a lesser extent than FGF19 in hepatocytes, while both FGF15 and FGF19 reduced *Cyp7a1* expression with almost equal potency [141,147]. These data suggest that FGF15 exists as a monomer or a dimer with a configuration quite different from FGF19.

### 4.4. Regulation of CYP7A1 Expression by FGF15/19

As described above, CYP7A1 is the critical enzyme for regulating BA production and is one of the most important targets of FGF15/19 signaling. However, the detailed mechanism by which FGF15/19 represses *Cyp7a1* levels in the liver is not completely understood. *Cyp7a1* transcription is maintained by two key nuclear receptors, the HNF4α homodimer and LRH-1 monomer, both of which show binding elements flanking each other on the 5′-upstream of the transcription initiation site [11,15,149]. In fact, the liver-specific disruption of *Hnf4a* and *Lrh-1* genes significantly reduced *Cyp7a1* expression levels [150]. Both FXR activation by BA and FGFR4 by FGF15/19 in hepatocytes results in SHP induction, an atypical nuclear receptor that lacks DBD and suppresses activity of their binding targets [151,152]. The promoter region of *Shp* has an IR1 FXR-RXRα binding element and *Shp* expression levels are induced following stimulation with BA, revealing that FXR is one of its principal regulators. Both intraperitoneal injection of FGF15/19 and oral administration of FXR agonist GW4064 strongly elevated *Shp* expression levels, whereas *Fgf15*-deficiency abolished only GW4064-dependent *Shp* elevation. These results suggest FGF15/19 themselves potently induce *Shp* expression without FXR activation [141]. ChIP data revealed that both HNF4α and LRH-1antibodies enriched the binding element on the *Cyp7a1* promoter from the liver chromatin and re-ChIP assays, in which chromatin in liver overexpressing epitope-tagged SHP was first immunoprecipitated with an LRH-1 antibody, then subjected to a second round of ChIP with an epitope antibody that also enriched the element. This data confirmed that SHP actually binds to HNF4α and LRH-1 to suppress their activity [150]. However, in addition to oral administration of GW4064 by *Shp*-deficient mice suppressing hepatic *Cyp7a1* levels, intraperitoneal FGF19 administration also suppressed *Cyp7a1* transcription levels without changing SHP protein levels in the liver overexpressing *Shp* [153]. These data strongly suggest the existence of SHP-independent machineries for FGF15/19-mediated *Cyp7a1* suppression. For example, JNK phosphorylation was induced in hepatocytes following FGF19 stimulation and is involved in SHP-independent *Cyp7a1* suppression [11,153]. Recent evidence further revealed that FGF15/19 inhibited nuclear translocation of transcription factor EB (TFEB), which binds to the promoter region of *Cyp7a1* to induce transcription by inducing phosphorylation via mTOR/ERK activation in an SHP-independent manner [154]. Collectively, FGF15/19-dependent and independent suppression of *Cyp7a1* levels may be regulated with more complicated machinery.

### 4.5. Regulation of Other Biological Events by FGF15/19

FGF15/19 induces different biochemical events via the FGFR4-KLβ heterodimer complex located in the plasma membrane. FGFR4 itself harbors an intracellular tyrosine kinase domain and elicits tyrosine phosphorylation on FRS2 upon FGF15/19 binding. Tyrosine-phosphorylated FRS2 is allowed to bind to growth factor-bound protein 2 (GRB2) to activate downstream signaling cascades to induce phosphorylation of ERK1/2 [155,156,157], after which phosphorylated ERK1/2 promotes mitogenesis of hepatocytes to regenerate liver tissue under physiological conditions [158,159,160] and progress hepatocarcinoma [128,161,162], neck squamous cell carcinoma [163], and gallbladder carcinoma cells [164] under pathogenic conditions. To eliminate these potentially harmful side effects, FGF19 analogs lacking mitogenic activity have been developed for future medication based on the aforementioned fact that some FGF19 mutants elicit normal *Cyp7a1* reduction in the liver, but do not induce mitogenesis.

FGF15/19 reduces transcription of metabolic enzymes by inhibiting the cAMP responsive element-binding protein (CREB)-peroxisome proliferator-activated receptor-gamma coactivator (PGC)-1α pathway [165] and inducing phosphorylation of p90 ribosomal S6 kinase to increase protein synthesis in the liver without inducing Akt or p70 ribosomal S6 kinase phosphorylation, which activates mechanistic targets of rapamycin (mTOR) signaling. FGF15/19 also induced phosphorylation of glycogen synthase kinase (GSK) 3α and GSK3β to suppress these activities and enhance glycogen synthesis in the liver [141,166]. For de novo lipogenesis, FGF19 inhibits hepatic lipogenesis by increasing STAT3 activity and decreasing PGC-1β expression to suppress sterol regulatory element-binding protein (SREBP)-1c activity [21]. FGF19 can inhibit SREBP-1c and 2 activities via increased SHP [21,167]. In addition, oral administration of GW4064 to WT mice induced hepatic expression of *Insig2*, a negative regulator of SREBP activity, whereas *Fgf15* deficiency abolished induction [141]. FGF19 levels can increase following bariatric surgery and this increase is involved in postoperative body weight loss [168]. Collectively, FGF15/19 positively impacts our health status through these intracellular events. In fact, FGF19 analogs are now considered promising medication for hepatic disorders, such as NASH and NAFLD, whereby NGM282 is currently part of clinical trials [169,170].

### 4.6. Non-Hepatic Targets of FGF15/19

Recent publications show FGF15/19 plays a role in non-hepatic tissues. The most extensively studied tissue is skeletal muscle, in which FGF19 stimulates ERK and S6 phosphorylation to induce hypertrophy [171]. FGF19 can also ameliorate sarcopenia and skeletal muscle atrophy induced by glucocorticoid treatment or obesity, suggesting that FGF19 and its analogs has potential therapeutic application for these symptoms. FGF19 is also considered to stimulate the FGFR4-KLβ complex in the central nervous system (CNS) to improve energy metabolism in addition to its role in neural development. Administering FGF19 into a liver-specific *Klb*-deficient mouse with diet-induced obesity can still significantly reduce body weight accompanied with lower food intake, reduced serum plasma concentration, and hepatic triglyceride levels [172,173]. Administering BA stimulated FGF15 secretion, leading to activated receptors and silencing hypothalamic AGRP/NPY neurons to improve glucose tolerance [174]. Furthermore, a transgenic mouse model of *FGF19* generated by hydrodynamic gene transfer technology showed that FGF19 increased water intake, which appeared to be mediated via CNS receptors [175]. FGF15 can be detected in neurons in the dorsomedial hypothalamus, where glucagon secretion is negatively controlled and reduced vagal nerve firing improves glucose tolerance [176,177]. However, a comprehensive analysis using a highly sensitive RNAscope in situ hybridization and droplet digital PCR technology failed to detect neurons expressing both *Fgfr4* and *Klb* in the mouse brain [178]. Detail analyses are necessary to understand how FGF15/19 stimulates neural cells to elicit these functions.

## 5. Perspective

Due to the benefits of the FXR-FGF15/19 signaling axis on human health, significant research effort has focused on developing medication to target this axis mainly for liver disease, such as NASH and NAFLD. We are currently expanding beyond basic research to unravel the underlying mechanisms of the FXR-FGF15/19 signaling axis for successful clinical usage of their agonists/antagonists, primarily for hepatic disorders. However, relatively less attention has been paid in understanding the relationship between FGF15/19-dependent and independent signaling of FXR in the liver. Regarding the effect of FXR agonists on the FGF15/19 expression, it is already known that oral administration of GW4064 increases *Shp* expression in the liver of WT mice, but not FGF15-deficient mice. Although GW4064 has been removed as a potential FXR agonist in future medical application, due to its low solubility to water, the results strongly suggest newly-developed FXR agonists similarly affect the liver, at least partly via FGF19 signaling in humans. Since endogenous FGF19 may unavoidably influence proliferation of hepatocarcinoma cells, using FXR agonists on patients for long term or in high dosages may result in serious side effects. Conversely, regarding FGF19 activity and its analogs in the liver, we do not have enough data to understand how these compounds suppress *Cyp7a1* and genes that facilitate hepatic disorder development. Unidentified biochemical and molecular biological events remain to be explored and, ultimately, the hope is we can unravel the detailed machinery of the BA-FXR-FGF15/19 signaling axis.

## Figures and Tables

**Figure 1 ijms-23-06046-f001:**
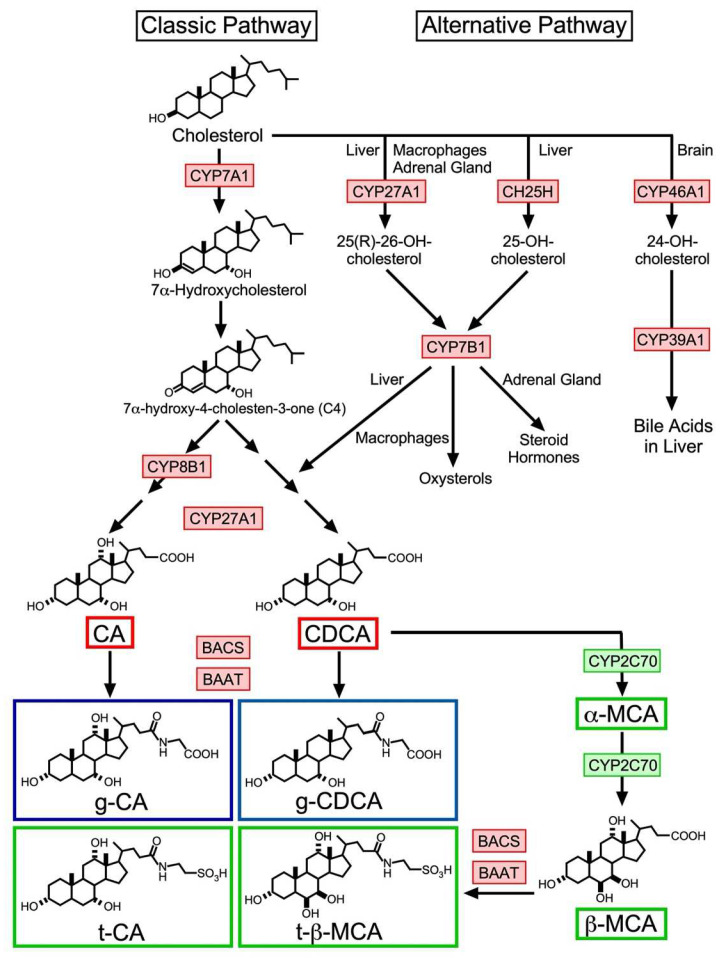
BA biosynthesis pathways in humans and mice. The classic bile acid biosynthesis pathway begins with converting cholesterol to 7α-hydroxycholesterol by CYP7A1 and subsequent conversion to C4. C4 is then converted to CA or CDCA by multiple enzymes including CYP8B1 and CYP27A1. CA and CDCA are then conjugated with taurine (t-) or glycine (g-) by BACS and BAAT. In mice, CDCA is further converted to α-MCA, then β-MCA by CYP2C70, which is not present in humans. Alternative pathways are initiated by CYP27A1 in the liver, macrophages, and the adrenal gland, cholesterol 25-hydroxylase (CH25H) in the liver and CYP46A1 in the brain. Products of the first two pathways are further converted by CYP7B1 to different steroidal compounds to produce BAs, oxysterols, and steroid hormones. The last pathway is involved in regulating cholesterol levels in the brain and the final product by CYP39A1 is delivered to the liver for BA synthesis. Red boxes indicate enzymes and BAs in both humans and mice, blue boxes indicate BAs mainly in humans, and green boxes indicate enzymes and BAs existing only in mice.

**Figure 2 ijms-23-06046-f002:**
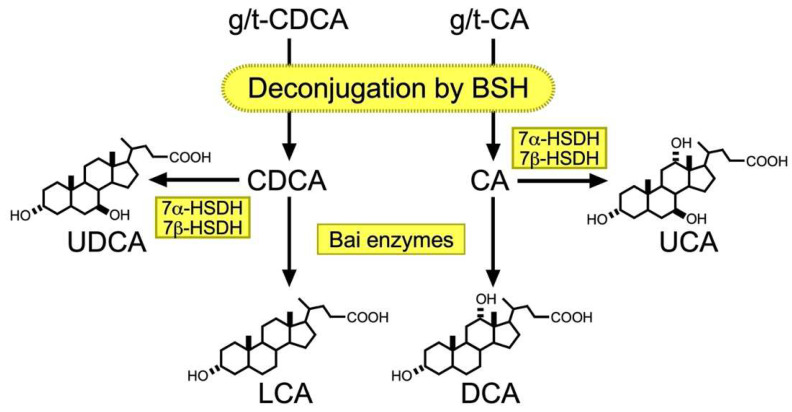
Secondary BA biosynthesis. Primary BAs are initially deconjugated by BSH, which is encoded by most gut microbiota. Deconjugated CDCA is then converted to UDCA and by 7α- and 7β-HSDH expressed in different bacteria or LCA by multiple steps of catalytic reactions by Bai enzymes in *Clostridium* spp. Deconjugated CA is also converted to UCA by 7α- and 7β-HSDH or DCA by Bai enzymes.

**Figure 3 ijms-23-06046-f003:**
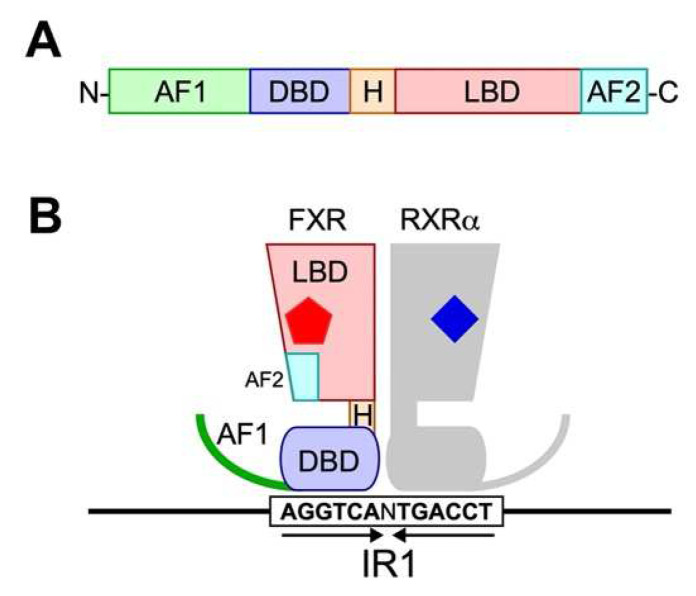
Protein structure of FXR. (**A**) General structure of nuclear receptors. A typical nuclear receptor consists of functional domains, such as activation of function 1 domain (AF1), DNA-binding domain (DBD), hinge region (H), ligand-binding domain (LBD), and activation of function 2 domain (AF-2). (**B**) Schematic representation of ligand-FXR-RXRα complex bound to IR1 element. Arrows indicate the inverted repeats of FXRE. Red pentagon and blue rhombus represent a bile acid molecule and an RXRα agonist, such as 9-cis-retinoic acid, respectively.

**Figure 4 ijms-23-06046-f004:**
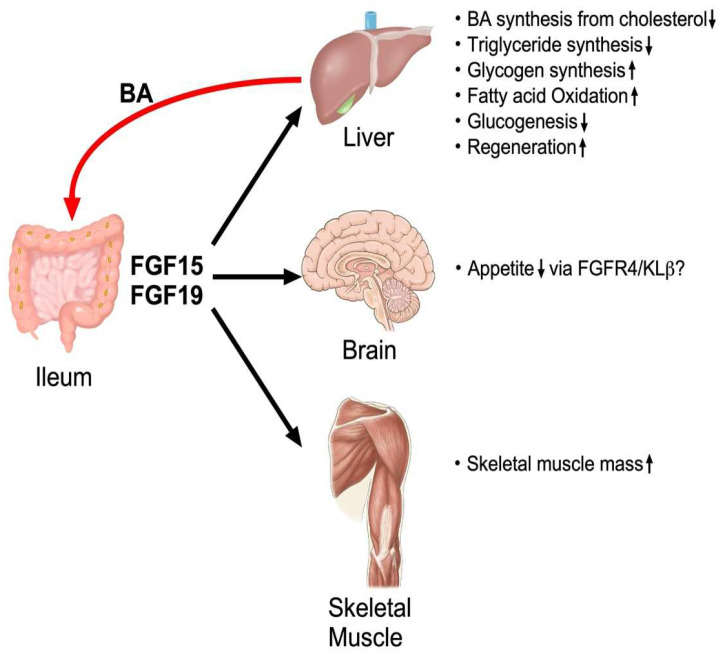
Overall physiological function of FGF15/19.

**Figure 5 ijms-23-06046-f005:**
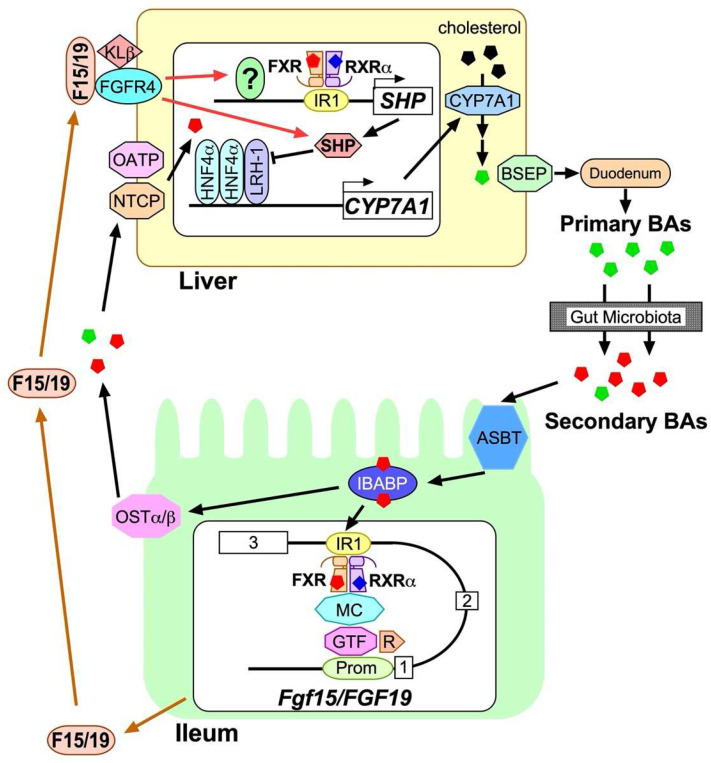
Regulation of BA synthesis by FXR-FGF15/19 signaling axis. Primary BAs (green pentagons) are synthesized from cholesterol (black pentagons) by a series of enzymatic reactions initiated with CYP7A1 in hepatocytes, and are exported from hepatocytes through BSEP, delivered to the duodenum, and converted to secondary BAs (red pentagons) by gut microbiota when traveling through the intestinal tract to help incorporation of fat-soluble nutrients. The rest of BAs are mostly reabsorbed through ASBT on the apical brush border of enterocytes in the ileum and bind to IBABP. The majority of BAs are then transported to the basolateral membrane to be exported into enterohepatic circulation, whereas a fraction of them is transported to the nucleus to form a complex with FXR and liganded RXRα on IR1 located between exon 2 and 3 of *Fgf15/FGF19*. The complex is then given access to general transcription factor (GTF) complexes on the promoter region (Prom), mediated by the mediator complex (MC), to initiate RNA polymerization by RNA polymerase II (R). Synthesized FGF15/19 (F15/19) is released into enterohepatic circulation and reaches the FGFR4-Klβ complex to activate intracellular events and induce *SHP* expression in hepatocytes for subsequent suppression of *CYP7A1* expression. Exported BAs into enterohepatic circulation are incorporated into hepatocytes through NTCP or OATP and bind to hepatic FXR-RXRα complex on IR1 of *SHP* to induce expression. The SHP protein then binds to the HNF4α homodier-LRH-1 complex to suppress CYP7A1 expression.

## Data Availability

Not applicable.

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
