# Peer review of "Molecular Basis of Bile Acid-FXR-FGF15/19 Signaling Axis"

_ijms, 2022, doi:10.3390/ijms23116046_

Round 1
Reviewer 1 Report
This review by Katafuchi and Makishima nicely summarizes current literature on bile acid metabolism, signaling, and involved molecular mechanisms. The authors provide sufficient references and discuss all important pathways and functions in a very compelling manner. There are only a few minor concerns which need to be changed:
Minor concerns:
- Page 2: please explain the abbreviation of FGF (only done in abstract so far) and CYP7A1 (done in row 76 but first mentioned in row 55)
- Please provide respective references for the first paragraph of BA biosynthesis
- Page 2, Row 74: Though clear to almost everyone, the authors should mention at least once that the primary substrate for BA synthesis is cholesterol, which is then further processed to 7a-hydroxcholesterol in the first step of the classical BA synthesis pathway
- Page 2, Row 80: HSD3B7 - its dehydrogenase, not hydrogenase
- Page 2, Row 88: correct term for BAAT would be “Bile acid-CoA:amino acid N-acyltransferase”
- Figure 1: Please indicate CYP27A1 in the classic pathway
- Page 3, Row 109: Please put the abbreviation CH25H, so also the figure is then easier to follow
- Page 3, Row 111: Is this correct? Cholesterol converted to (25R)-25-hydroxycholesterol (but in Figure its only 25-OH-cholesterol)? Please re-check and stay uniform with the nomenclature.
- Page 4: Can the authors be more specific on brain cholesterol clearance? Is cholesterol directly packed into lipoproteins (HDL, LDL) in brain cells or is this happening within the circulation?!
- Figure 2: Is the scheme not true for t-CDCA?
- Page 4, Row 151: CA for DCA, but CDCA for LCA!
- Page 5: 2.3 should be 2.4
- Page 6, Row 222: Please indicate (H) for hinge region, refer in Row 223 to Figure 3A
- Figure 3: Please indicate N- and C-terminal end of the structure; explain what is IR1 element?
- Figure 3B should me mentioned in the text as well (e.g. in line 242)
- Page 7, Row 276: The authors should rephrase the sentence to “In the kidney,” and not “in the latter two tissues”, as description of kidney tubules has nothing to do with adrenal glands
- Figure 4 is a bit misleading as it seems like BA from the liver enter the ileum. Though reabsorbed by the ileum, BAs enter the intestine in the duodenum. It would be appreciated if the figure is adapted accordingly.
- Page 8: Please provide references for the introduction paragraph on FGF15/19
- Figure legend 5, Row 341: black pentagons instead of “filled”; space too much in rows 348, 351, 354, 355?
- Page 10, Row 377: β-Klotho was already abbreviated on Page 2!
- Page 10, Row 407: Can the authors briefly describe what VSL#3 probiotics are? As antibiotics treatment resulted in greatly reduced ileal Fgf15 expression, one might expect the opposite effect of probiotics treatment, but the authors state also reduced Fgf15 expression after probiotics administration
- Page 12, Row 475: Please explain abbreviations of FRS2 and GRB2
Reviewer 2 Report
This is a comprehensive review of the Farnesoid-X-Receptor and the Retinoid-X-Receptor, their mode of action, as well as their agonists and of putative therapeutic targets. The Farnesoid-X-Receptor and the Retinoid-X-Receptor form a heterodimer, which is activated by bile acids, and in turn regulates gene expression involved in the metabolism of cholesterol, bile acids and lipids, mainly involving the liver. The expression of fibroblast growth factor (FGF) 15/FGF19 is identified as being regulated by the heterodimer, which in turn activates hepatic FGF receptor 4 -klotho receptor complex. It is this complex that regulates gene expression involved in the metabolism of above-mentioned biologicals and as well as cell proliferation. The possibility of agonists of the Farnesoid-X-Receptor and analogues of FGF15/19 as therapeutic targets for metabolic syndrome and cholestatic diseases are discussed.
GENERAL COMMENTS
The review is worthwhile of publication. The paper is well written and provides a clear overview of a complicated field. The reference list is extensive.
SPECIFIC COMMENTS
Ln 213: ‘…not completely characterized, significant differences were observed in tissue distribution…’
Author Response
Dear reviewer #2,
We appreciate giving us your helpful comment. We corrected the sentence accordingly.
Regards,
Reviewer 3 Report
Bile acids (Bas) binds to Farnesoid-X-Receptor (FXR) to induce FGF15/19 expressions in ileum, then FGF15/19 activate FGFR4/β-Klotho to regulate BA synthesis in liver. The feedback loop for BA synthesis can be served as pharmacological targets. The authors in this review article elucidated “Molecular Basis of Bile Acid-FXR-FGF15/19 Signaling Axis”. It is well-written and interesting review article in the field. The authors are suggested to cite the reference (Maliha et al., Farnesoid X receptor and fibroblast growth factor 15/19 as pharmacological targets, Liver Research, 2021) and English edit is needed for publication.
Author Response
Dear reviewer #3,
We appreciate giving us your helpful comments. We cited the suggested paper in the rows 302 and 320 (reference #108, Maliha et al. Liver Res. 2021). We also read the manuscript carefully again and corrected some grammatically incorrect usages of words. Furthermore, this manuscript has received the proofreading by molecular biology/biochemistry experts who speak English as a primary language.
Regards,